# Using micro-CT to explore bone density variations in the skulls of the vulnerable *Opsariichthys uncirostris uncirostris* (Three-lips fish) during reproductive migration to a Lake Biwa tributary

**Andrew Mvula**[1]*, **Daisuke Tawara**[2]°, **Atsushi Maruyama**[1]°

1 Department of Environmental Solution Technology, Faculty of Science and Technology, Ryukoku University, Seta, Japan, 2 Department of Mechanical Engineering and Robotics, Faculty of Science and Technology, Ryukoku University, Seta, Japan

° These authors contributed equally to this work.
* andrew.mvula@yahoo.com

**Data Availability Statement:** All relevant data are within the manuscript and its Supporting Information files.

## Abstract

Not much is known about the changes in bone density due to fish reproductive migration. We used micro-CT and inferential statistics to determine whether the relative bone density in the skulls of adult Three-lips fish, that seasonally upstream migrated to a Lake Biwa tributary, changed across their known reproductive season. The relative bone density significantly decreased as standard length and condition factor (*K*) increased in both sexes. This negative relationship is likely due to age and hormonal effects in the fish. Results from the bone density analysis also revealed that male Three-lips fish had potentially lower relative bone density (although not significantly different) than females during peak reproductive migration, i.e., July to August. On average, male Three-lips fish are larger in length and weight than females, and in many species, females prefer larger males to smaller males, viewing their size as an indicator of genetic fitness and their ability to provide protection. Resources in the skulls of Three-lips males may be distributed in such a way that increases reproductive success, i.e., size at the expense of quality. In addition, individuals with slightly less dense bones, particularly males, appeared later than those with denser bones during the peak of the reproductive season. The high energy demands involved with aggression in males, often requires resource mobilization from various tissue compartments and could explain the slightly lower density in the latter half of the peak migration. Furthermore, Three-lips individuals that migrated earlier and later during the reproductive season may have more energy reserves than those that had been in the river for some time, hence the variable bone density between individuals. This study serves as a foundation for future studies on the effects of migration, changes in physiology and age on bone density analysis of Three-lips fish and other species in various ecosystems.

**Funding:** This research was supported by grants from the 2021-2022 the Joint Research Center for Science and Technology of Ryukoku University as Japan Society for the Promotion of Science (JSPS) KAKENHI (Grants Nos. 16K00630, 18KK0208 and 23KK0131). The funders did not play any role in the study design, data collection and analysis, decision to publish, or preparation of the manuscript.

**Competing interests:** The authors have declared that no competing interests exist.

# Introduction

Fish, such as the small teleost *Danio rerio* (zebrafish), which rely mostly on their skeletal architecture for structural support and mineral homeostasis, have been widely used to model skeletal morphogenesis in human skeletons due to similarities in developmental mechanisms [1, 2]. Although the skeletal system primarily informs function, feeding and mobility adaptations, the assessment of the fish skeletal system, such as bone density, can also offer a unique perspective in determining the intricacies of fish physiology, and provide insights into the health status of individuals [2]. The skeletogenesis of fish is progressive, often involving mineralization and compositional modulation of skeletal tissues [3]. Such a process reflects the interplay between environmental factors and endogenous physiological processes [4]. Integrating bone density analysis in the study of the potamodromous fish has potential to unveil a unique dimension linked to the reproductive migration of the species. Potamodromy, characterized by fish that complete their life cycles within freshwater systems, often involves extensive upstream migrations for spawning purposes [5, 6].

The relationship between bone density and reproductive migration has ecological significance. Changes in bone density can offer insights into the energetic demands and physiological adaptations associated with migratory endeavors [7]. As potamodromous species navigate challenging aquatic environments during reproductive migrations, alterations in bone density may serve as indicators of the metabolic investments required for successful migration and subsequent reproduction [2, 7]. A number of tools can be used to evaluate this relationship including, biochemical markers (e.g., C-terminal telopeptide), nutritional analyses, hormonal analyses and computed tomography [8]. Selecting the right tools for evaluation not only enhances our comprehension of the life history strategies of potamodromous fish but also provides a valuable tool for assessing the impact of anthropogenic activities on critical migratory corridors, ultimately contributing to the conservation and sustainable management of these essential fish populations [3, 4]. Understanding of bone density variations in fish, especially between sexes, can also provide insights into species-specific adaptations related to reproductive behavior [9]. The environment and dietary factors can affect nutrient recruitment in tissues, such as bones, and have an impact on the overall health of the fish [3].

Conventional health measurements like Fulton's condition constant (*K*), even though widely used, can not adequately identify drastic changes in fish energetics especially during the reproductive season when weight changes can be observed due to reproductive behavior [10, 11]. By assessing bone density, however, it should be possible to identify sudden changes in the health or energetics of migrating populations [12]. The newly developed technique micro-focus computed tomography (micro-CT), coupled with calcium hydroxyapatite (CaHA) phantoms, provides a powerful tool for studying bone density in fish [13]. Unlike conventional methods of assessing bone density, e.g., chemical methods that may alter the structure and composition of bones, micro-CT provides high-resolution three-dimensional imaging of bone structures, allowing for detailed quantitative analysis of parameters without altering the structure or composition of the bones [14–16]. The inclusion of CaHA phantoms in micro-CT protocols promotes standardization by mimicking the x-ray attenuation properties of bones, therefore enabling calibration of CT images to bone density [17]. This calibration is crucial for accurate and reliable measurements of bone density [13, 17]. The non-destructive nature of micro-CT also allows us to examine precious specimens and make informed decisions on the status of internal structures with minimal damage to the specimen [18].

*Opsariichthys uncirostris uncirostris* (Three-lips fish), an endemic yet vulnerable potamodromous fish that relies on lake–river migration for its reproductive migration in the Lake Biwa ecosystem, migrates and initiates feeding at different times in Lake Biwa tributaries [19].

Three-lips fish and its related species in Asia, which lack dental teeth, have evolved unique jaws to catch fish within the constraints of the cyprinid family [20]. Although the cost of migration has been discussed in evolutionary ecology of migratory fishes, not much is known about the changes in bone density related to reproductive migration. Therefore, this research aimed at exploring the bone density variations in the skulls of the vulnerable potamodromous Three-lips fish by using micro-CT derived bone density. The research questions for this study were as follows: is the bone density of the cranium in Three-lips fish related to size and health of the fish? Are there differences in the bone density of the carnium in Three-lips individuals sampled in different months during their reproductive season? Are there bone density differences in the cranium of males and females during their reproductive migration?

## Materials and methods

### Description of study site

The samples in this study were collected in the lower reaches of the Shiotsuo River (under the control of the Shiga Prefectural local government), which flows into Lake Biwa from the north. The river has a total length of 9 km and a basin area spanning 21.8 m$^2$ [21]. The river flows through mountainous terrain (95.8% of the total river catchment area), contributing to a steep gradient of 9.5 m/km, making it one of Lake Biwa's steepest rivers and a preferred reproductive upstream migration route for Three-lips fish when they seasonally migrate from May to September. Reproductive migration of Three-lips fish in Lake Biwa has been documented in several studies using conventional and environmental DNA (eDNA) analysis [22, 23]. A study on $\delta^{13}$C and $\delta^{15}$N stable isotopes in muscle and mucus tissues of Three-lips fish during their reproductive migration to the Shiotsuo River also indicates that the fish may migrate and inhabit spawning sites at varying times, with some individuals staying longer than others in the river [19]. Due to its steep gradient, the river is fast-flowing, and it is also characterized by gravel bottom substrates, all of which have been identified to be key drivers of Three-lips reproductive migration [22]. In addition to the ease of sample collection, Shiotsuo River is perennial, and there are no high weirs in the middle and lower reaches of the river, allowing for natural upstream migration of Three-lips fish. These characteristics make Shiotsuo river a conducive environment for conducting bone density studies on Three-lips fish during its reproductive upstream migration.

It is expected that Three-lips fish caught at different times during their reproductive migration in Shiotsuo River will have varying bone density in their cranium due to size and health, time the species was caught as well as sex of the individual.

### Fish sampling and biometric measurements

Three-lips fish samples were collected monthly in the lower reach (2–3 km from the river mouth) of Shiotsuo River in 2019 during their reproductive migration from May to September using cast nets. Fish sampling in the Shiotsuo River was done with approval from the Department of Fisheries of Shiga Prefecture (permit number: 30–34, flag numbers: 274, 454 and 537). The sampling plan was systematic and a target of 20 Three-lips individuals was set for each month (i.e., 20 individuals over 5 months, in total of 100). Despite sampling efforts, only 57 individuals were collected from the planned 100 (Table 1).

The fish were sacrificed by placing them in ice water and in compliance with Japanese laws and standards. Since the fish were caught during their reproductive migration, no control specimens were obtained. Wet weight (g) and standard length (mm) were measured (to the nearest 0.01 g or 0.1 mm) in the field. The sex of the fish was confirmed in the field by dissecting the fish and checking the gonads for sperm and eggs. Fish were put in individual Ziplock

**Table 1. Number of Three-lips fish individuals (males and females) collected in Shiotsuo River each sampling month.**

| Sampling Month | Number of Three-lips individuals collected | | Sex ratio (males to females) |
|---|---|---|---|
| | **Males** | **Females** | |
| May | 0 | 0 | N/A |
| June | 11 | 1 | 11:1 |
| July | 10 | 10 | 1:1 |
| August | 9 | 11 | 9:11 |
| Seprember | 3 | 2 | 3:2 |

bags and kept on ice in a cooler box while in the field. After the field sampling, samples were transported to the laboratory at Ryukoku University and kept frozen below –22.5°C until analysis. The Fulton's condition constant (*K*) was calculated for each fish according to the following formula:

$$K = 100 \times \frac{W}{L^3}$$

Where: *W* (g) is the wet weight of the fish and *L* (cm) is the standard length of the fish.

## Sample preparation and micro-computed tomography

Prior to micro-CT scanning, the samples were placed in 70% ethanol for at least 48 hrs. to facilitate water removal. Next, the samples were cut along the dorsoventral axis just before the pelvic fin. This was done to ensure that the samples fit into a scanning container and field of view of the micro-CT scanner. The samples were then washed in absolute ethanol to remove any impurities on the surface of the fish and fixed in the scanning container together with a Micro-CT-HA phantom (QRM, Möhrendorf, Germany) containing 5 rods of Calcium hydroxyapatite (CaHA) with known densities (0, 50, 200, 800, 1200 mg CaHA/cm³, respectively). The phantom was secured to the sample by means of Sellotape® (Fig 1A). All samples were scanned at 70 kV and 40 μA with a 1024 × 1024 resolution and YZ smoothing using an inspeXio SMX-100CT Micro Focus X-Ray CT System (Shimadzu Corporation, Kyoto, Japan). The slice size and voxel size were 0.079 ± 0.009 (mean ± SD) and 0.041 ± 0.043 (mean ± SD), respectively. The scan files were then exported as 8-bit bitmap files (Fig 1B) for subsequent processing in 3D slicer (a 3D imaging freeware) [24].

## Measuring relative bone density in skulls of Three-lips fish

The cranium (skull bones) in scans were segmented (until the preopercle) using Otsu Thresholding (pixel intensity thresholding range: 60–255) due to its simplicity and speed in the Segmentation editor module of 3D slicer (Fig 1C–1E, [25]). The average pixel intensity, i.e., the specific area of the scan that has an accompanying value of x-ray absorption strength (on a grayscale, weak absorption is dark (black or 0 grayscale value) and strong absorption is bright (white or 255 grayscale value); [26]), for the segmentation on the cranium, superimposed on the original scan, was then recorded using the Statistics function of the Segmentation module of 3D slicer (Fig 1F). Similary, the average pixel intensities for each of the 5 phantom rods were obtained and used to create calibration curves (in the slope-intercept form) for converting pixel intensity of the scans to bone density (mg CaHA/cm³). The $R^2$ value in all phantom calibrations was 0.99 (S1 Fig).

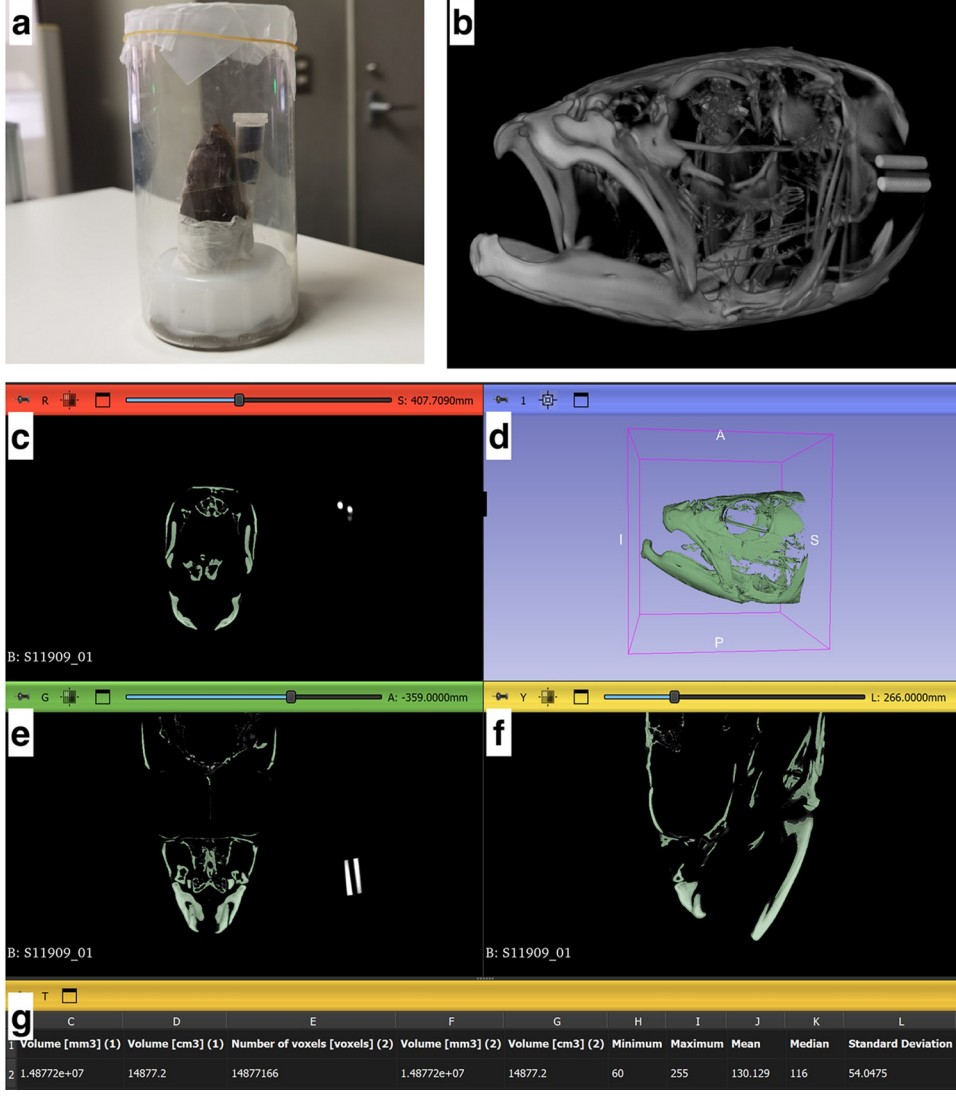

**Fig 1. Workflow for assessing relative bone density using micro-CT.** (a) The sample is placed in a scanning container with CaHA phantoms. (b) Scanned images are exported as 8-bit bitmap files for processing in 3D Slicer. (c–f) Bones are segmented using Otsu thresholding (pixel intensity range 60–255) and unwanted regions are removed using the scissors tool in 3D Slicer. (g) The pixel intensity in the segmented area, superimposed on the original scan, is recorded using the statistics function in 3D Slicer.

## Data analysis and interpretation

All data analyses were conducted in R ver. 4.3.1 software [27]. The relationship between the biometrics standard length (mm), sex and condition factor ($K$), as explanatory variables, and relative bone density (as a response variable) were assessed using a generalized linear model (GLM) with a guassian family [base::glm() in R]. Variable selection for the final model was then achieved through forward and reverse stepAIC [base::stepAIC() in R].

The asessment of bone density trends across the sampling period was done in two steps. First, the non-parametric Kruskal-Wallis test [base::kruskal.test() in R] was used to assess whether the observed relative bone density trends, with the interaction between sex and sampling month as an explanatory variable, were significant. If the Kruskal-Wallis test was significant, then a multiple comparison test using the Steel-Dwass test was conducted using the

NSM3::pSDCFlig() with "Monte Carlo" as a method in the function (due to the relatively small sample size in this study). The advantage of using the Steel-Dwass test over conventional methods, such as the Dunnet method, is in its ability to solve multiple comparison problems more easily [28]. When statistically comparing the relative bone density between males and females of Three-lips fish, only datasets from the July catch ($n = 10$ males, $n = 10$ females) and August catch ($n = 9$ males, $n = 11$ females) were used due to sample size limitations. The sample sizes in the June catch ($n = 11$ males, $n = 1$ female) and September catch ($n = 3$ males, $n = 2$ females) were not large enough to perform statistical comparisons. In parallel, the analysis was done on male individuals only comparing the June catch ($n = 11$ males), July catch ($n = 10$ males), and August catch ($n = 9$ males) due to a sufficient sample size for statistical analysis.

## Results

### Relationship between biometrics and micro-CT obtained bone density

In both males and female Three-lips fish, bone density decreased with increasing standard length and condition factor ($K$) (Fig 2A and 2B). The GLM model with gaussian family and stepAIC revealed that the relative bone density decreased as the explanatory variables standard length, condition factor increased and a tendency for lower bone density in males when compared to females (Table 2). The standard length and condition factor ($K$) had significant effects on the relative bone density in the GLM ($p < 0.05$) while sex did not have a significant effect on the relative bone density ($p > 0.05$). The negative relationship of condition factor and bone density was contrary to the expectation that healthier individuals, regardless of size, would have a larger bone density than slightly less healthier individuals. It is worth mentioning that the changes in standard length and condition factor ($K$) of Three-lips fish across different months have already been documented by Mvula and Maruyama [19]. In their paper, both the standard length and condition factor decreased during peak reproductive migration (July to August), which was attributed to environmental changes, rising water temperatures, and potential alterations in food availability. To effectively and statistically assess the effects of monthly variations in standard length and condition factor ($K$) on bone density, a larger sample size than the one currently obtained for this study is required.

### Bone density distribution across the reproductive season

There was a significant difference in bone density between the July and August catches (Kruskal-Wallis $X^2 = 12.34$, $d.f. = 3$, $p < 0.05$). Bone density was lower in the August catch when compared to the July catch for both males and females (Fig 3). Although bone density was higher in females than in males between the July and August catches, pairwise comparison revelead that the differences were only significant between females caught in July and males caught in August ($W\ statistic = 4.39$, $p < 0.05$). In males only, the bone density was lower for each sampling month when compared to its previous month except in September when the bone density was higher than the previous month. However, the Kruskal-Wallis test revealed that the observed differences in relative bone density were not significantly different from each other. These results indicate that there may be differences in bone density due to sex, however, a bigger sample size is need to confirm this observation.

## Discussion

### Effect of age and environment on bone density

In this study, the relative bone density in Three-lips fish decreased as standard length and condition factor increased in both sexes (Fig 2, Table 2). As species grow, the rate at which

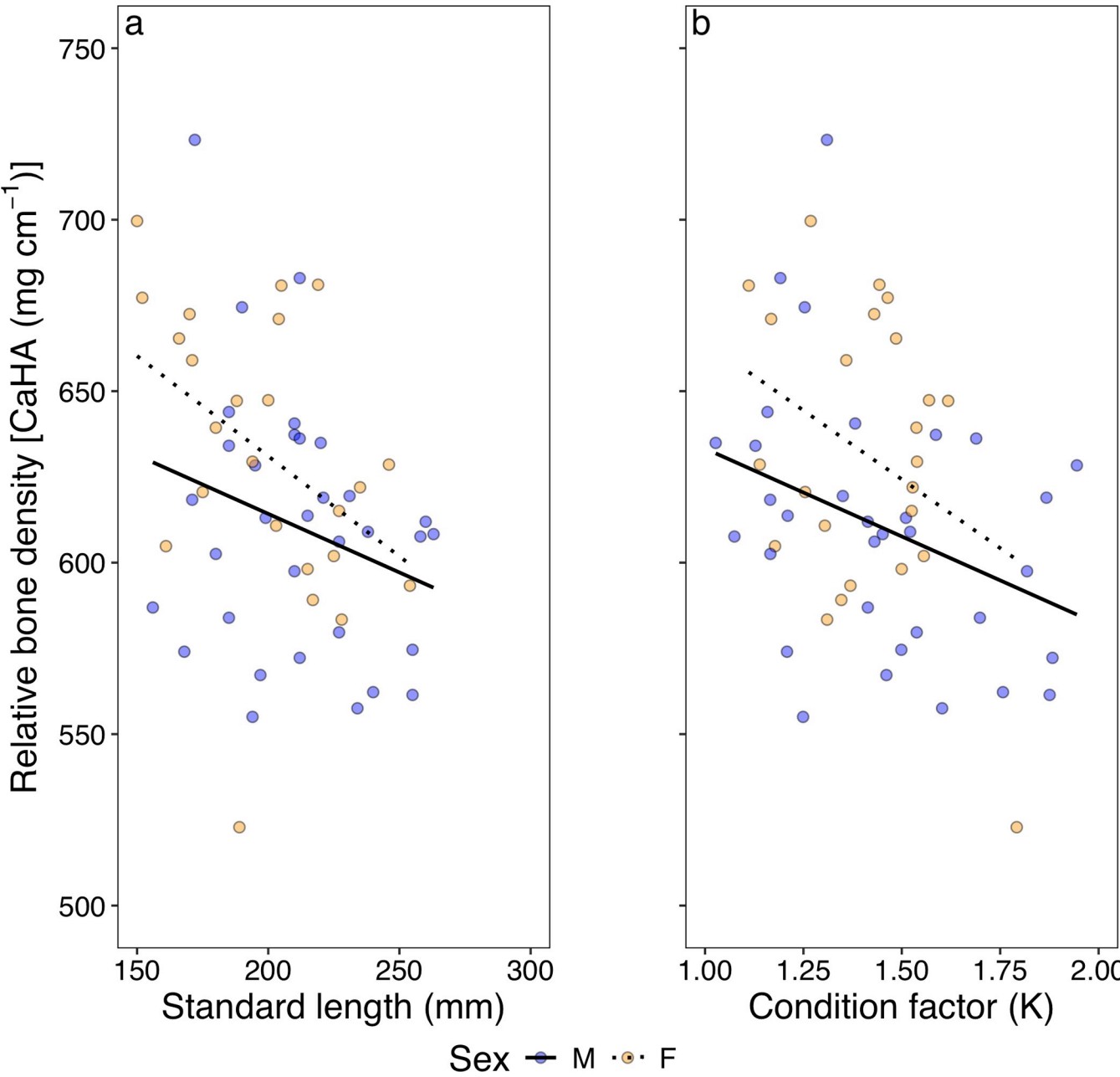

**Fig 2.** Relationship between relative bone density and (a) standard length and (b) condition factor. Blue and orange points represent male and female Three-lips fish, respectively. Black solid and dotted lines indicate the linear fit for males and females, respectively.

materials, especially calcium hydroxyapatite, replenish in bones decreases probably due to age and hormonal effects in the fish [13, 29]. However, in the case of Three-lips fish, it is highly likely that the observed decline is due to hormonal effects, since migrating individuals during the reproductive seasonfall within the same older age group [30–32]. Fish bones play an important role in resource mobilization by acting as reserves for important nutrients. These nutrients, e.g., calcium hydroxyapatite, lipids, and proteins, are not only important for structural integrity of the skeletal system but may also act as an energy source for the fish in a stressed environment [2, 13]. Larger individuals require more energy to move their large

**Table 2. Coefficients on relative bone density evaluated in relation to biometrics (standard length, condition factor and sex) as explanatory variables using general linear models (GLMs) with a gaussian family selected via stepAIC.**

| Explanatory variables | Coefficients (estimates ± standard errors) | | |
|---|---|---|---|
| (Intercept) | 778.56 | ± | 42.20*** |
| Standard length (mm) | −0.38 | ± | 0.17* |
| Condition factor (K) | −50.23 | ± | 21.63* |
| Sex[a] | −14.47 | ± | 9.88 |

Significance levels

(***$p < 0.001$

**$p < 0.01$

*$p < 0.05$).

[a] The estimate for "Sex" in the linear regression model reflects the effect on the dependent variable specifically for males compared to females.

bodies [33]. This could explain the slightly lower bone density in larger individuals. In addition, one would expect healthier individuals to exhibit higher densities compared to less healthier individuals. Even though all migrating individuals were healthy ($K > 1$), the negative relationship between condition factor ($K$) and bone density suggests some sort of resource investment strategy by Three-lips fish during their reproductive migration, perhaps to other tissue such as gonads which play a more direct role in spawning [34, 35]. Further research is required to ascertain resource utilization strategies from various tissues in Three-lips fish.

## Bone density distribution across the Three-lips migrating period

Assessing the Three-lips fish bone density distribution across different months during their reproductive migration helps identify potential variations in bone density that may be influenced by seasonal changes or migration patterns. In particular, the higher bone density in June and September when fewer individuals are migrating could be a result of one of three scenarios: lack of competition between individuals, early and late migrations within the Three-lips fish population, or a response to strenuous reproductive activities. First, using stable isotope analysis and catch data, Mvula and Maruyama [19] demonstrated tentative differences in the timing of upstream migration from Lake Biwa to Shiotsuo River with potential early and late migrations within the population. It is possible that due to fewer fish in the river, there is reduced competition for mates and consequently the absence of high energy demanding activities like chasing other individuals [36, 37]. Alternatively, it is possible that the higher bone density in June and September is because of new individuals entering the reproductive migration sites. There are possible differences in the onset of feeding and consequently upstream migration between individuals [19]. It is likely that these individuals, during the early and late migration waves, have enough energy reserves in tissues other than bones to undertake upstream reproductive migration. As such, no drastic changes in bone density are expected. Finally, the higher bone density could be due to bone remodeling after strenuous reproductive activity (e.g., the upstream swim and spawning). According to Wolff's Law on bone remodeling, the structure (and consequently density) of bone tissue in healthy individuals will adjust in response to the mechanical forces and stresses applied to it [38].

Bone remodeling is a complex process that involves the conversion of mechanical signals into biochemical signals in cellular signaling [39]. There is evidence of mechanically induced bone remodeling in the jaws of teleost fish, such as cichlids [40]. As such, the mechanical demands of reproductive activities could induce remodeling response in the skulls of Three-

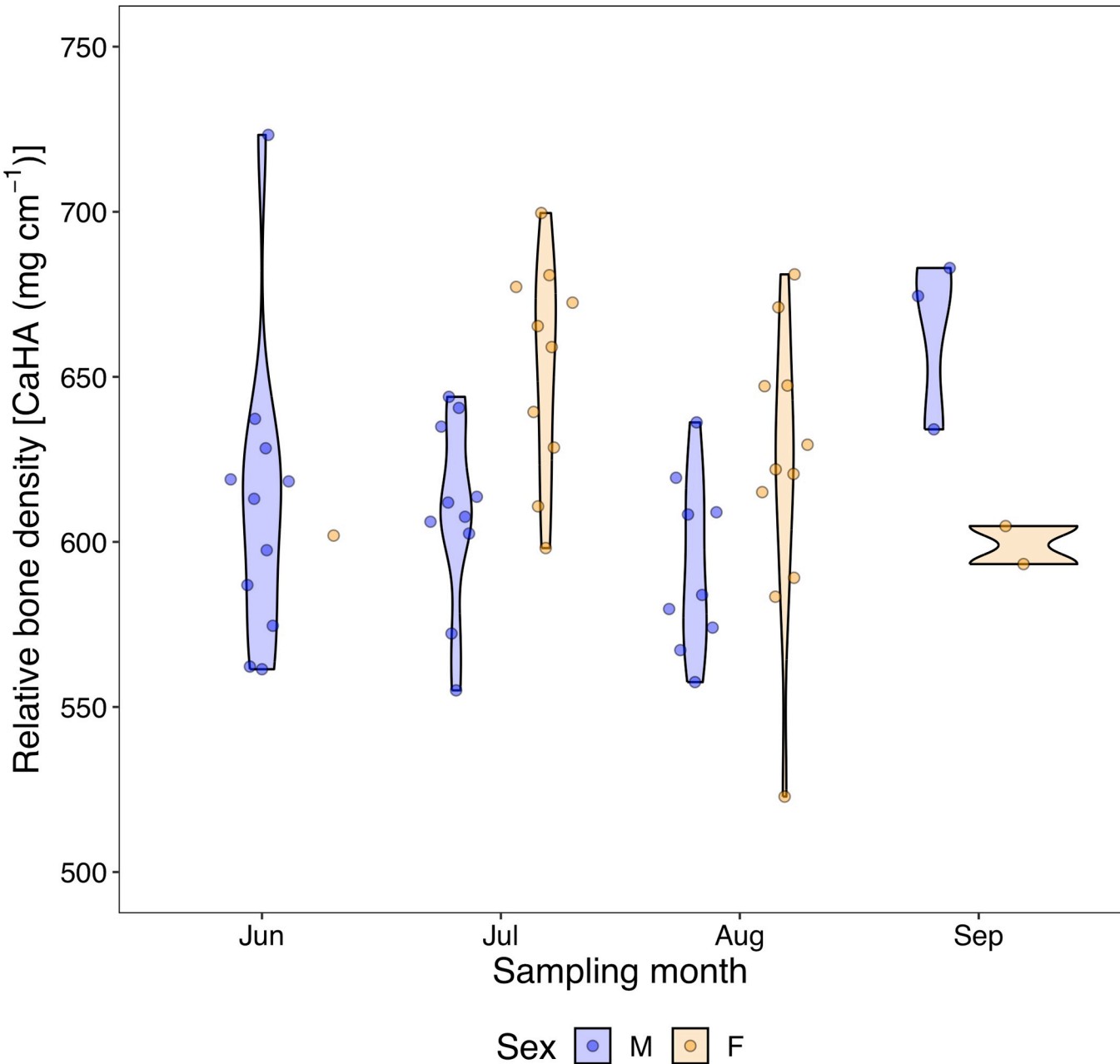

**Fig 3. Relative bone density [CaHA (mg cm⁻¹)] of Three-lips fish across the reproductive season.** Blue and orange violin plots (with points) represent male and female Three-lips fish, respectively. The width of the violin plots indicates the distribution and density of the data. Data points and violin plots are jittered for alignment. May data is excluded due to no fish being caught despite sampling efforts.

lips fish to cope with this stress. However, there is a need for further investigations on the behaviors of Three-lips fish before and after the peak migration period to draw conclusions on which scenario is more likely. For example, future studies may consider exploring the immediate impact of various stimuli (e.g., mechanical and chemical stimuli) on bone density in Three-lips fish. Furthermore, the bone density analysis in this study was done on the cranium as a whole, however, future studies should assess bone density differences in individual bones of the cranium. For example, these studies could focus on the differences in bone density

between the dentaries and premaxilla. This would not only provide insights into the dynamic nature of Three-lips fish bone remodeling in response to varying environments but could also inspire the development of new biotechnological approaches for bone tissue engineering and regeneration in humans, as is the case with zebrafish and other marine species [1, 41].

### Sex roles potentially influence bone density in Three-lips fish

The results suggested differences between male and female Three-lips fish with males having a relatively lower bone density than females (Figs 2 and 3). Besides the hormonal differences between males and females, male Three-lips fish are, on average, larger in length and weight [30]. The effect of size and its role in reproduction might help explain the lower bone density in males than in females. A larger head could be a consequence of evolutionary processes that make males appear more threatening to other species, thus appealing to females during reproductive migration. Generally, females in many species prefer larger males, viewing their size as an indicator of their ability to provide protection [42–44]. In addition, large sizes may be an indicator of good health and genetic fitness in individuals [45]. Therefore, resources in the skulls of Three-lips males may be distributed in such a way that increases reproductive success, i.e., size at the expense of quality. In addition, individuals with slightly less dense bones, particularly males, appeared later than those with denser bones during the peak of the reproductive season (around July-August). Three-lips fish, especially males, are known to be aggressive, often chasing and biting other males in the same area [46]. Having denser bones during the peak of the reproductive season (around July-August), when more females are migrating, could provide a better reproductive advantage, not only as a weapon but also during resource mobilization. However, a larger sample size is required to confidently conclude on the effects of sex on bone density in Three-lips fish.

In conclusion, this study demonstrated variations in bone densities in Three-lips fish of different ages and health during their reproductive migration to a Lake Biwa tributary. Although not significantly different, the study also highlighted possible differences between male and female Three-lips fish. We recognize the limitations in this study and that the results must be treated with caution due to the relatively low sample size. Nevertheless, the findings from this study serve as a foundation for ecologists and biomechanics hoping to study the effects of migration, changes in physiology and age on bone density changes in Three-lips fish and other species in various ecosystems.

## Supporting information

**S1 Fig. Calibration curve for converting greyscale value to relative bone density using a MicroCT-HA phantom (QRM, Möhrendorf, Germany).**
(TIF)

**S1 File. Raw data underlying the findings in this study.**
(CSV)

**S2 File. R script underlying the analysis and figures in this study.**
(PDF)

## Acknowledgments

We thank Dr. H. Sawada (Nishinihon Institute of Technology) for the assistance rendered during field sampling. We would also like to thank Dr. F. Ziadi (Okinawa Institute of Science and Technology) for the advice on micro-CT scanning. We express our gratitude to Dr. R. Tabata

and his team at the Lake Biwa Museum for their guidance and support during the Three-lips preliminary survey aquarium experiments. We would also like to thank the Shiga Prefectural Government for giving us with permission to collect fish samples in Shiotsuo River.

## Author Contributions

**Conceptualization:** Andrew Mvula, Daisuke Tawara, Atsushi Maruyama.

**Data curation:** Andrew Mvula.

**Formal analysis:** Andrew Mvula.

**Funding acquisition:** Daisuke Tawara, Atsushi Maruyama.

**Investigation:** Andrew Mvula.

**Methodology:** Andrew Mvula.

**Software:** Andrew Mvula.

**Supervision:** Daisuke Tawara, Atsushi Maruyama.

**Validation:** Andrew Mvula.

**Visualization:** Andrew Mvula.

**Writing – original draft:** Andrew Mvula.

**Writing – review & editing:** Andrew Mvula, Daisuke Tawara, Atsushi Maruyama.

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
