## [Decision Letter · Decision Letter 0]

22 May 2024

PONE-D-24-06945Using micro-CT to explore bone density variations in the skulls of the vulnerable Opsariichthys uncirostris uncirostris (Three-lips fish) during reproductive migration to a Lake Biwa tributaryPLOS ONE

Dear Dr. Mvula,

Thank you for submitting your manuscript to PLOS ONE. After careful consideration, we feel that it has merit but does not fully meet PLOS ONE’s publication criteria as it currently stands. Therefore, we invite you to submit a revised version of the manuscript that addresses the points raised during the review process.

This manuscript represents an interesting application of microCT to study of physiology and life history in wild fish population. The strength of the paper is that it is focused and concise, with few issues.

There are a few items that need the attention of the authors, all have to do with writing and interpretation.

Major points

1.    Note PLOS-ONE has strict data sharping policy. Data have to be made accessible. Use data.dryad.org, Figshare or comparable.

2.    The experimental design is a bit unclear, explain better when the fish are migrating, and if only migrating fish were caught (no controls?). This could be iterated in the results section, update statements about monhts with points about when migrating and when not.

3.    The influence of sex is not very convincing. This was not supported by the linear models. Check the results and dicussion from line 228 onwards. You can retain the discussion on potential differences, but don’t overstate the results.

4.    The discussion should be split into three sections, add a section on the month differences (migrating period), and move the section on “putative sex differences” down. See also abstract and conclusions.

We look forward to receiving your revised manuscript.

Kind regards,

Arnar Palsson, Ph.D.

Academic Editor

PLOS ONE

Journal Requirements:

"This research was supported by grants from the 2021-2022 the Joint Research Center for Science and Technology of Ryukoku University as Japan Society for the Promotion of Science (JSPS) KAKENHI (Grants Nos. 16K00630, 18KK0208 and 23KK0131)."

4. In the online submission form, you indicated that "The data underlying the findings of this study may be made available upon reasonable request from the corresponding author."

**Additional Editor Comments:**

This manuscript represents an interesting application of microCT to study of physiology and life history in wild fish population. The strength of the paper is that it is focused and concise, with few issues.

There are a few items that need the attention of the authors, all have to do with writing and interpretation.

Major points

1. Note PLOS-ONE has strict data sharping policy. Data have to be made accessible. Use data.dryad.org, Figshare or comparable.

2. The experimental design is a bit unclear, explain better when the fish are migrating, and if only migrating fish were caught (no controls?). This could be iterated in the results section, update statements about monhts with points about when migrating and when not.

3. The influence of sex is not very convincing. This was not supported by the linear models. Check the results and dicussion from line 228 onwards. You can retain the discussion on potential differences, but don’t overstate the results.

4. The discussion should be split into three sections, add a section on the month differences (migrating period), and move the section on “putative sex differences” down. See also abstract and conclusions.

Intermediate points

Line 51. “The assessment of the fish skeletal system, such as bone density, not only offers a unique perspective in determining the intricacies of fish physiology, and but also provides insights into the health status of individuals”

I would say that the skeletal system primarlity informs about function, feeding and mobility adaptations, physiology and health are then secondary.

Line 65

“subsequent reproduction [2, 7].” Are there other ways to evaluate this relationship? Mention complementary methods, here or in discussion.

Line 96-

Clarify the aims of study, and provide one or more specific research questions or hypotheses.

Lines 106-115.

Description of system is lengthy. What in this is really needed for this study, the RQ, results and discussion? These fish migrate, is there seasonality in that or not??

Lines 185-

Clarify description, the “negative relationship” applies to the quantatitaive variables, but does not hold meaning for the sex differences. Split into two sentences, say explicitly Sex did not have a sign. effect. Also, not necessary to use the word “biometrics”, rather use explanatory variable or similar. Also the expectation in line 187, are bigger fish thought to be more healthy?

Line 205

Your overall model did not support this, but still you conclude…

“These results indicate that females have higher densities when compared to males during the reproductive season.” I don’t think the data support this conclusion. Rather that the sex effects were small if real, bigger sample needed to confirm. Use in discussion also.

Line 210-

Frame the discussion better. Mention the organism, and iterate why comparison of months is important. It is a bit unclear if all the fish are migrating, or if only fish in some months are migrating??

Line 265.

Other future studies could also be envisioned. Can you propose a replicate of this study, that can more thoroughly answer the research questions? More sampling, telemetry, isotopes ect???

Conclusions, Line 272 -

“In conclusion, this study demonstrated variations in bone densities between male and female” not supported strongly by data. Start with the result most strongly supported.

Also the statement “the distinct changes in bone density during each catch are likely because of resource mobilization strategies employed by Three-lips fish to ensure successful reproduction.

Can this be concluded from the data? Build this argument better in discussion!

Concluding sentence could be more direct, “about the influence of migration, changes in physiology etc on bone density variation”…

Minor points

Line 56.

Drop mention of “this study”, make the statement more general.

Line 72

Add “of”

“Understanding of bone density variations in fish”

Line 76 and 77

“drastic” used twice. Replace or drop.

Line 79

micro-CT is not novel, maybe “newly developed”? When was this developed??

Line 83

“Images” not “imaging”

Line 89

Drop “allows us to”

Line 104

Study was not completed in the river, the samples came from the river. Rewrite.

Line 123

Each month? Better than sampling, also give total sampled in study “(total XX fish)”.

Line 133

Replace “After the 48 hrs. had elapsed,” with “Next”

Line 179

Simplify to

“sufficient sample size for”

Line 191.

Drop “(estimates ± standard errors)” from table title.

Also, the estimate between sexes, is that male minus female or other way around? Explain in either title or footnote.

Line 200

Perhaps “pairwise comparison revealed” instead of “the comparisson test revelead”??

Line 215.

Drop “are”?? “reproductive season are fall within the”

Line 219

Could rather than would

Line 222

Drop “to” in front of “the negative relationship between”

Line 224

Drop “muscle” - “gonads” is the main deal?

Line 246

Reword “migration” …”fewer fish are migrating could” – and drop “as”

Line 247

Drop “likely”

Line 249

Citing the figure does not make sense here, it only shows data, and does not explain these models.

Line 249

“due to the reduced numbers,” what does this refer to? Fewer fish?

Lome 252

Reword “new individuals approaching the”, “new individuals enter the”

Line 255

Reword, not sure what this means “journeys and this is not reflected in the changes in bone”

Line 432. Fig2 legend.

Simplify to

“Relationship between relative bone density and (a) standard length and (b) condition factor.”

And

“Solid lines indicate the linear fit for the relationships”.

Reviewers' comments:

Reviewer's Responses to Questions

**Comments to the Author**

1. Is the manuscript technically sound, and do the data support the conclusions?

Reviewer #1: Yes

2. Has the statistical analysis been performed appropriately and rigorously? 

Reviewer #1: Yes

3. Have the authors made all data underlying the findings in their manuscript fully available?

Reviewer #1: Yes

4. Is the manuscript presented in an intelligible fashion and written in standard English?

Reviewer #1: Yes

5. Review Comments to the Author

Reviewer #1: My recommendation is to accept the article with minor revisions:

Minor comments:

Methods:

1) how were fish sacrificed in the field? Did this comply with permitting standards?

2) How did you determine sex in the three lips fish?

3) please identify which bones in the skull were used in the density measurements. Are there density differences at the level of individual bones or is the density similar across all facial bones?

4) are there bone size differences in addition to density differences?

Line 53: “skeletogenesis of fish is a progressive” should read “skeletogenesis of fish is progressive…”

Line 143: cite 3D Slicer: Fedorov A., Beichel R., Kalpathy-Cramer J., Finet J., Fillion-Robin J-C., Pujol S., Bauer C., Jennings D., Fennessy F.M., Sonka M., Buatti J., Aylward S.R., Miller J.V., Pieper S., Kikinis R. 3D Slicer as an Image Computing Platform for the Quantitative Imaging Network. Magnetic Resonance Imaging. 2012 Nov;30(9):1323-41. PMID: 22770690. PMCID: PMC3466397.

Line 214: remove “are” from the sentence.

6. PLOS authors have the option to publish the peer review history of their article (what does this mean?). If published, this will include your full peer review and any attached files.

Reviewer #1: No

---

## [Author Response · Author response to Decision Letter 0]

5 Jul 2024

Dear Prof. Arnar Palsson,

We wish to submit a revised version of our original research article entitled “Using micro-CT to explore bone density variations in the skulls of the vulnerable Opsariichthys uncirostris uncirostris (Three-lips fish) during reproductive migration to a Lake Biwa tributary” for consideration by the Plos One journal.

Re: Resubmission of revised manuscript No. PONE-D-24-06945

Thank you for inviting us to submit a revised draft of our original manuscript entitled, “Using micro-CT to explore bone density variations in the skulls of the vulnerable Opsariichthys uncirostris uncirostris (Three-lips fish) during reproductive migration to a Lake Biwa tributary” to the PLOS ONE journal. 

We would like to express our sincere gratitude to you, the reviewers and the editor for the time and effort dedicated to reviewing the manuscript. The valuable feedback has greatly contributed to improving the quality of the manuscript.

Revisions in the manuscript are shown as track changes (line numbers mentioned are from the revised manuscript with all mark-up and revisions inline). Overall, we have formatted the manuscript according to the PLOS ONE templates. The data underlying this study has also been included as supplementary material to the submission (including R script). We would also like to clarify that "The funders had no role in study design, data collection and analysis, decision to publish, or preparation of the manuscript." The reference list has also been reviewed and two new references [8, 23, 25] have been added to the manuscript after revesion. Finally, the figures have also been checked using the PACE digital diagnostic tool. Below this letter we address each of the concerns raised by you, the reviewers and other editors.

Thank you once again for considering our manuscript for publication in the Plos One journal. We look forward to your feedback on the revised version and hope for a favorable outcome.

Sincerely,

Andrew Mvula

Responses to specific concerns

1.00 Journal Requirements

Concern 1.01: Please ensure that your manuscript meets PLOS ONE's style requirements, including those for file naming. 

Response 1.01: The manuscript has been modified in accordance to PLOS ONE’s style requirements provided in the templates.

Concern 1.02: Please note that PLOS ONE has specific guidelines on code sharing for submissions in which author-generated code underpins the findings in the manuscript. In these cases, all author-generated code must be made available without restrictions upon publication of the work.

Response 1.02: The R code underlying the findings in the manuscript has been shared as supporting information (S2_File.R).

Concern 1.03: Please state what role the funders took in the study. If the funders had no role, please state: "The funders had no role in study design, data collection and analysis, decision to publish, or preparation of the manuscript."

Response 1.03: The role of the funders has been mentioned in the rebuttal letter as follows: “The funders had no role in study design, data collection and analysis, decision to publish, or preparation of the manuscript.” 

Concern 1.04: In the online submission form, you indicated that "The data underlying the findings of this study may be made available upon reasonable request from the corresponding author."

Response 1.04: The data underlying this study (S1_File.csv) has been provided as supplemtary material in the submission. 

Concern 1.05: Please review your reference list to ensure that it is complete and correct. If you have cited papers that have been retracted, please include the rationale for doing so in the manuscript text, or remove these references and replace them with relevant current references. Any changes to the reference list should be mentioned in the rebuttal letter that accompanies your revised manuscript. If you need to cite a retracted article, indicate the article’s retracted status in the References list and also include a citation and full reference for the retraction notice.

Response 1.05: The references have been checked and ensured it is complete and correct. Where necessary, any changes have been mentioned in the rebuttal letter.

2.00 Major points

Concern 2.01: Note PLOS-ONE has strict data sharping policy. Data have to be made accessible. Use data.dryad.org, Figshare or comparable.

Response 2.01: The data underlying the findings in this study (S1_File.csv) has been attached as supporting material to the submission.

Concern 2.02: The experimental design is a bit unclear, explain better when the fish are migrating, and if only migrating fish were caught (no controls?). This could be iterated in the results section, update statements about monhts with points about when migrating and when not.

Response 2.02: The period when the fish are migrating has been clarified in the materials and methods section as follows: from L144 (revised manuscript with track changes) “…a preferred reproductive upstream migration route for Three-lips fish when they seasonally migrate from May to September. Reproductive migration of Three-lips fish in Lake Biwa has been documented in several studies using conventional and environmental DNA (eDNA) analysis [22-23]. A study on δ13C and δ15N stable isotopes in muscle and mucus tissues of Three-lips fish during their reproductive migration also indicates that the fish may migrate and inhabit spawning sites at varying times, with some individuals staying longer than others in the Shiotsuo River [19].”. In addition, a statement on controls was added as follows: from L154 (revised manuscript with track changes) “Since the fish were caught during their reproductive migration, no control specimens were obtained.”

Concern 2.03: The influence of sex is not very convincing. This was not supported by the linear models. Check the results and dicussion from line 228 onwards. You can retain the discussion on potential differences, but don’t overstate the results.

Response 2.03: The section on influence of sex has been modified to not overstate results in the “Sex roles potentially influence bone density in Three-lips fish” section as follows: from L307 (revised manuscript with track changes) “Results from the bone density analysis also indicated that there are potential differences between male Three-lips fish and female Three-lips fish with males having a relatively lower bone density than females (Figs 2-3)... However, a larger sample size is required to confidently conclude on the effects of sex on bone density in Three-lips fish.”

Concern 2.04: The discussion should be split into three sections, add a section on the month differences (migrating period), and move the section on “putative sex differences” down. See also abstract and conclusions.

Response 2.04: The discussion has been split into three sections as suggested: i) effect of age and environment on bone density, ii) bone density distribution across the Three-lips migrating period and iii) sex roles potentially influence bone density in Three-lips fish.

3.00 Intermediate points

Concern 3.01: Line 51. “The assessment of the fish skeletal system, such as bone density, not only offers a unique perspective in determining the intricacies of fish physiology, and but also provides insights into the health status of individuals”I would say that the skeletal system primarlity informs about function, feeding and mobility adaptations, physiology and health are then secondary.

Response 3.01: The statement has been modified as follows: from L59 (revised manuscript with track changes) “Although the skeletal system primarily informs function, feeding and mobility adaptations, the assessment of the fish skeletal system, such as bone density, can also offer a unique perspective in determining the intricacies of fish physiology, and provide insights into the health status of individuals.”

Concern 3.02: Line 65 “subsequent reproduction [2, 7].” Are there other ways to evaluate this relationship? Mention complementary methods, here or in discussion.

Response 3.02: The complementary methods have been mentioned in the introduction section as follows: from L75 (revised manuscript with track changes) “A number of tools can be used to evaluate this relationship including, biochemical markers (e.g., C-terminal telopeptide), nutritional analyses, hormonal analyses and computed tomography. Selecting the right tools for evaluation not only enhances our comprehension of the life history strategies of potamodromous fish but also provides a valuable tool for assessing the impact of anthropogenic activities on critical migratory corridors, ultimately contributing to the conservation and sustainable management of these essential fish populations…”

Concern 3.03: Line 96-Clarify the aims of study, and provide one or more specific research questions or hypotheses.

Response 3.03: The specific research questions have been stated as follows: from L112 (revised manuscript with track changes) “The specific research question for this study were as follows: is the bone density of the cranium in Three-lips fish related to size and health of the fish? Are there differences in the bone density of the carnium in Three-lips individuals sampled in different months during their reproductive season? Are there bone density differences in the carnium of male and female Three-lips individuals sampled during their reproductive migration?”

Concern 3.04: Lines 106-115. Description of system is lengthy. What in this is really needed for this study, the RQ, results and discussion? These fish migrate, is there seasonality in that or not??

Response 3.04: We feel it is important to describe the syetem in detail because it paints a picture of the reproductive migration journey the Three-lips fish undertakes. On the other concerns, the element of seasonality has been clarified as follows: from L124 (revised manuscript with track changes) “…a preferred reproductive upstream migration route for Three-lips fish when they seasonally migrate from May to September. Reproductive migration of Three-lips fish in Lake Biwa has been documented in several studies using conventional and environmental DNA (eDNA) analysis [22-23]. A study on δ13C and δ15N stable isotopes in muscle and mucus tissues of Three-lips fish during their reproductive migration also indicates that the fish may migrate and inhabit spawning sites at varying times, with some individuals staying longer than others in the Shiotsuo River [19].” The expected results have also been stated as follows: from L139 (revised manuscript with track changes) “It is expected that Three-lips fish caught at different times during their reproductive migration in Shiotsuo River will have varying bone density in their cranium due to size and health, time the species was caught as well as sex of the individual.”

Concern 3.05: Lines 185- Clarify description, the “negative relationship” applies to the quantatitaive variables, but does not hold meaning for the sex differences. Split into two sentences, say explicitly Sex did not have a sign. effect. Also, not necessary to use the word “biometrics”, rather use explanatory variable or similar. Also the expectation in line 187, are bigger fish thought to be more healthy?

Response 3.05: The statements have been modified as follows to address the concerns: from L219 (revised manuscript with track changes) “The GLM model with gaussian family and stepAIC revealed that the relative bone density decreased as the explanatory variables standard length, condition factor increased and a tendency for lower bone density in males when compared to females (Table 1). The standard length and condition factor (K) had significant effects on the relative bone density in the GLM (p < 0.05) while sex did not have a significant effect on the relative bone density (p > 0.05). The result of the GLM on condition factor (K) was contrary to the expectation that healthier individuals, regardless of size, would have a larger bone density than slightly less healthier individuals.” 

Concern 3.06: Line 205. Your overall model did not support this, but still you conclude… “These results indicate that females have higher densities when compared to males during the reproductive season.” I don’t think the data support this conclusion. Rather that the sex effects were small if real, bigger sample needed to confirm. Use in discussion also.

Response 3.06: The statement has been modified as follows: from L247 (revised manuscript with track changes) “These results indicate that there may be differences in bone density due to sex, however, a bigger sample size is need to confirm this observation.”

Concern 3.07: Line 210- Frame the discussion better. Mention the organism, and iterate why comparison of months is important. It is a bit unclear if all the fish are migrating, or if only fish in some months are migrating??

Response 3.07: The discussion section has been modified to reflect the concerns. In particular the discussion section has been split into three sections as follows: i) effect f age and environment on bone density, ii) bone density distribution across the Three-lips migrating period and iii) sex roles potentially influence bone density in Three-lips fish. The organism has also been mentioned. In addition, a previous study, that clarifies if or some fish are migrating (using stable isotopes) has been cited as follows: from L127 (revised manuscript with track changes) “A study on δ13C and δ15N stable isotopes in muscle and mucus tissues of Three-lips fish during their reproductive migration also indicates that the fish may migrate and inhabit spawning sites at varying times, with some individuals staying longer than others in the Shiotsuo River [19].”

Concern 3.08: Line 265. Other future studies could also be envisioned. Can you propose a replicate of this study, that can more thoroughly answer the research questions? More sampling, telemetry, isotopes ect???

Response 3.08: The following studies has been envisioned: from L295 (revised manuscript with track changes) “However, there is a need for further investigations on the behaviors of Three-lips fish before and after the peak migration period to draw conclusions on which scenario is more likely. For example, future studies may consider exploring the immediate impact of various stimuli (e.g., mechanical and chemical stimuli) on bone density in Three-lips fish. Furthermore, the bone density analysis in this study was done on the cranium as a whole, however, future studies should assess bone density differences in individual bones of the cranium. For example, these studies could focus on the differences in bone density between the dentaries and premaxilla. This would not only provide insights into the dynamic nature of Three-lips fish bone remodeling in response to varying environments but could also inspire the development of new biotechnological approaches for bone tissue engineering and regeneration in humans, as is the case with zebrafish and other marine species [1, 41].”

Concern 3.09: Conclusions, Line 272 -“In conclusion, this study demonstrated variations in bone densities between male and female” not supported strongly by data. Start with the result most strongly supported. Also the statement “the distinct changes in bone density during each catch are likely because of resource mobilization strategies employed by Three-lips fish to ensure successful reproduction. Can this be concluded from the data? Build this argument better in discussion! Concluding sentence could be more direct, “about the influence of migration, changes in physiology etc on bone density variation”…

Response 3.09: The conclusion has been modified as follows: from L355 (revised manuscript with track changes) “In conclusion, this study demonstrated variations in bone densities in Three-lips fish of different ages and health during their reproductive migration to a Lake Biwa tributary. Although not significantly different, the study also highlighted possible differences between male and female Three-lips fish, with females potentially having higher bone density than males. We recognize the limitations in this study and that the results must be treated with caution due to the relatively low sample

---

## [Editor Report · Decision Letter 1]

28 Aug 2024

PONE-D-24-06945R1Using micro-CT to explore bone density variations in the skulls of the vulnerable Opsariichthys uncirostris uncirostris (Three-lips fish) during reproductive migration to a Lake Biwa tributaryPLOS ONE

Dear Dr. Mvula,

Thank you for submitting your manuscript to PLOS ONE. After careful consideration, we feel that it has merit but does not fully meet PLOS ONE’s publication criteria as it currently stands. Therefore, we invite you to submit a revised version of the manuscript that addresses the points raised during the review process.

The manuscript is mostly in good shape.

A key question is whether the correlated variables (length and conditition factor) also differ by months? Please provide a similar analyses as in Figure 3 for the correlates. Are these month differences in bone density are driven by size or condition variation?

Also, use consistent colours to indicate the sexes in different figures. Consolidate and standardize legends and caption texts about the graphs also.

See notes for minor points.

We look forward to receiving your revised manuscript.

Kind regards,

Arnar Palsson, Ph.D.

Academic Editor

PLOS ONE

Journal Requirements:

**Additional Editor Comments:**

PONE_Ctmulva

The manuscript is mostly in good shape.

A key question is whether the correlated variables (length and conditition factor) also differ by months? Please provide a similar analyses as in Figure 3 for the correlates. Are these month differences in bone density are driven by size or condition variation?

Also, use consistent colours to indicate the sexes in different figures. Consolidate and standardize legends and caption texts about the graphs also.

Minor points

Line 63

Drop “progressive“ Such a progressive process

Line 70.

Tone down bombastic wording “The interplay between bone density and reproductive migration holds profound ecological significance.”

Line 88

Split “cannot”

Line 90

Drop “the” in front of bone densitiy

Line 112

Make plural (multiple questions), drop “specific”. “The specific research question for this”

Line 116

Reword „“carnium of male and female Three-lips individuals sampled during” to “cranium of males and females during”

Line 130

“with some individuals staying longer than others in the Shiotsuo River“ Clarify, is this the river they spawn in?

Line135

This reference to Aye fish is not needed - drop “...as well as Ayu fish. Ayu fish serves as the primary food source for Three-lips fish in the Lake Biwa ecosystem [198, 2432].”

Line 148

Refer to this as a “plan”. Reword “20 individuals x 5 months to collect a total of” to “20 individuals over 5 months, in total 100“

Line 150-153

Shorten “Despite… ” And refer to a table about sample sizes, sex ratio etc. Use info from line 208- can simplify that section also.

Line 180

Drop “ the head“

Line 193

Add reference to R

Line 211

Replace “However…” with, “In parallel, the analyses was done one male..”

Line 225

Reword „“The result of the GLM on condition factor (K)is” to “The negative relationship of condition factor and bone density was contrary“

Line 234

Drop “using a dummy variable approach”

Line 271

This opening sentence has circular logic. Fix.

Line 277-78

Reword. The word “potential” is used to often

Line 308

Reword “ from the bone density analysis also indicated that there are potential differences “ to “The results suggested differences “

Line 311

Drop “ than females “

Line 358

Drop “ with females potentially having higher bone density than males”

---

## [Author Response · Author response to Decision Letter 1]

1 Sep 2024

Dear Prof. Arnar Palsson,

We wish to submit a revised version of our original research article entitled “Using micro-CT to explore bone density variations in the skulls of the vulnerable Opsariichthys uncirostris uncirostris (Three-lips fish) during reproductive migration to a Lake Biwa tributary” for consideration by the Plos One journal.

Re: Resubmission of revised manuscript No. PONE-D-24-06945R1

Thank you for inviting us to submit a revised draft of our original manuscript entitled, “Using micro-CT to explore bone density variations in the skulls of the vulnerable Opsariichthys uncirostris uncirostris (Three-lips fish) during reproductive migration to a Lake Biwa tributary” to the PLOS ONE journal. 

We would like to express our sincere gratitude to you, the reviewers and the editor for the time and effort dedicated to reviewing the manuscript. The valuable feedback continues to improves the clarity and quality of the manuscript.

Revisions in the manuscript are shown as track changes (line numbers and pages mentioned are from the revised manuscript with all mark-up and revisions inline). Overall, we have also explained why the correlated variables (standard length and condition factor) were not assessed monthly in this study. We have also used consistent colors to indicate sexes in the different figures. The reference list has also been reviewed and a new reference for the R programming software has been added. Finally, grammatical errors have also been addressed. Below this letter we address each of the concerns raised by you, the reviewers and other editors. 

Thank you once again for considering our manuscript for publication in the Plos One journal. We look forward to your feedback on the revised version and hope for a favorable outcome.

Sincerely,

Andrew Mvula

Responses to specific concerns

1.0 Journal requirements

Concern 1.01: Please review your reference list to ensure that it is complete and correct.

Response 1.01: The reference list has been reviewed and ensured it is complete and correct.

2.00 Key concerns

Concern 2.01: A key question is whether the correlated variables (length and conditition factor) also differ by months? Please provide a similar analyses as in Figure 3 for the correlates. Are these month differences in bone density are driven by size or condition variation?

Response 2.01: Regarding the differences between length and condition factor across different months has already been assessed using the same biometric data as well as stable isotopes by Mvula & Maruyama 2024. A brief explanation on these changes has been added in L.215 p.11 with reference to the paper as follows “It is worth mentioning that the changes in standard length and condition factor (K) of Three-lips fish across different months have already been documented by Mvula and Maruyama [19]. In their paper, both the standard length and condition factor decreased during peak reproductive migration (July to August), which was attributed to environmental changes, rising water temperatures, and potential alterations in food availability. To effectively and statistically assess the effects of monthly variations in standard length and condition factor (K) on bone density, a larger sample size than the one currently obtained for this study is required.”. As to whether the differences in bone density are driven by size or condition variation, this has already been shown in Table 2 using linear modelling but to effectively assess the monthly differences affect bone density then a larger sample size than currently obtained for this study is required.

Concern 2.02: Also, use consistent colours to indicate the sexes in different figures. Consolidate and standardize legends and caption texts about the graphs also.

Response 2.02: Colors in the figures have been revised and are consitent for the sexes. The caption and legend texts have also been standardized.

3.00 Minor points

Concern 3.01: Line 63. Drop “progressive“ Such a progressive process

Response 3.01: L.51 P.4, Progressive has been dropped from the statement.

Concern 3.02: Line 70. Tone down bombastic wording “The interplay between bone density and reproductive migration holds profound ecological significance.”

Response 3.02: L.58 P.4, The statement has been revised as follows: “The relationship between bone density and reproductive migration has ecological significance.”

Concern 3.03: Line 88. Split “cannot”

Response 3.03: L.75 P.5, the word “cannot” has been split to “can not”.

Concern 3.04: Line 90. Drop “the” in front of bone densitiy

Response 3.04: L.77 P.5, the has been dropped in front of bone density.

Concern 3.05: Line 112. Make plural (multiple questions), drop “specific”. “The specific research question for this”

Response 3.05: L.99 P.6, the statement has been revised as follows “The research questions for this study were as follows…”

Concern 3.06: Line 116.Reword „“carnium of male and female Three-lips individuals sampled during” to “cranium of males and females during”

Response 3.06: L.102 P.6, The statement has been reworded as follows “Are there bone density differences in the cranium of males and females during their reproductive migration?”

Concern 3.07: Line 130.“with some individuals staying longer than others in the Shiotsuo River“ Clarify, is this the river they spawn in?

Response 3.07: L.116 p.7, the statement has been revised as follows “A study on δ13C and δ15N stable isotopes in muscle and mucus tissues of Three-lips fish during their reproductive migration to the Shiotsuo River also indicates that the fish may migrate and inhabit spawning sites at varying times, with some individuals staying longer than others in the river”

Concern 3.08: Line135.This reference to Aye fish is not needed - drop “...as well as Ayu fish. Ayu fish serves as the primary food source for Three-lips fish in the Lake Biwa ecosystem [198, 2432].”

Response 3.08: L.122 p.7, the reference to Ayu fish has been dropped from the statement.

Concern 3.09: Line 148.Refer to this as a “plan”. Reword “20 individuals x 5 months to collect a total of” to “20 individuals over 5 months, in total 100“

Response 3.09: L.134 p.7, the statement has been revised as follows “The sampling plan was systematic and a target of 20 Three-lips individuals was set for each month (i.e., 20 individuals over 5 months, in total of 100).”

Concern 3.10: Line 150-153. Shorten “Despite… ” And refer to a table about sample sizes, sex ratio etc. Use info from line 208- can simplify that section also.

Response 3.10: L.136 p.8, the statement has been revised as follows “Despite sampling efforts, only 57 individuals were collected from the planned 100 (Table 1).” A table has also been added.

Concern 3.11: Line 180.Drop “ the head“

Response 3.11: L.169 p.9, “the head” has been dropped from the statement.

Concern 3.12: Line 193. Add reference to R

Response 3.12: L.182 p.10, The reference has been added to R.

Concern 3.13: Line 211. Replace “However…” with, “In parallel, the analyses was done one male..”

Response 3.13: L.200 p.11, the statement has been modified as follows “In parallel, the analysis was done on male individuals only comparing…”

Concern 3.14: Line 225. Reword „“The result of the GLM on condition factor (K)is” to “The negative relationship of condition factor and bone density was contrary“

Response 3.14: L.213 p.11, the statement has bee reworded as suggested.

Concern 3.15: Line 234. Drop “using a dummy variable approach”

Response 3.15: L.229 p.12, the statement has been dropped from the statement.

Concern 3.16: Line 271. This opening sentence has circular logic. Fix.

Response 3.16: L.266 p.14, the statement has been modified as follows “Assessing the Three-lips fish bone density distribution across different months during their reproductive migration helps identify potential variations in bone density that may be influenced by seasonal changes or migration patterns.”

Concern 3.17: Line 277-78. Reword. The word “potential” is used to often

Response 3.17: L.272 p.14, the statement has been reworded as follows “…demonstrated tentative differences in the timing of upstream migration from Lake Biwa to Shiotsuo River with potential early and late migrations within the population.”

Concern 3.18: Line 308. Reword “ from the bone density analysis also indicated that there are potential differences “ to “The results suggested differences “

Response 3.18: L.303 p.15, the statement has been reworeded as follows, “The results suggested differences between male and female Three-lips fish with males having a relatively lower bone density than females”

Concern 3.19: Line 311.Drop “ than females “

Response 3.19: L.2306 p.16, “than females” has been dropped from the statement.

Concern 3.20: Line 358. Drop “ with females potentially having higher bone density than males”

Response 3.20: L.325 p.16, the statement has been dropped.

---

## [Editor Report · Decision Letter 2]

2 Sep 2024

Using micro-CT to explore bone density variations in the skulls of the vulnerable Opsariichthys uncirostris uncirostris (Three-lips fish) during reproductive migration to a Lake Biwa tributary

PONE-D-24-06945R2

Dear Dr. Mvula,

We’re pleased to inform you that your manuscript has been judged scientifically suitable for publication and will be formally accepted for publication once it meets all outstanding technical requirements.

Kind regards,

Arnar Palsson, Ph.D.

Academic Editor

PLOS ONE
---

## [Editor Report · Acceptance letter]

8 Sep 2024

PONE-D-24-06945R2 

PLOS ONE

Dear Dr. Mvula, 

I'm pleased to inform you that your manuscript has been deemed suitable for publication in PLOS ONE. Congratulations! Your manuscript is now being handed over to our production team.

Kind regards, 

on behalf of

Dr. Arnar Palsson 

Academic Editor

PLOS ONE